# Memory Modulation by Exercise in Young Adults Is Related to Lactate and Not Affected by Sex or BDNF Polymorphism

**DOI:** 10.3390/biology11101541

**Published:** 2022-10-20

**Authors:** Juan Arturo Ballester-Ferrer, Alba Roldan, Eduardo Cervelló, Diego Pastor

**Affiliations:** Sports Research Centre, Department of Sport Sciences, Miguel Hernández University of Elche, Avda. de la Universidad s/n, 03202 Alicante, Spain

**Keywords:** cognitive function, BDNF, physical exercise, intensity, blood lactate

## Abstract

**Simple Summary:**

The health and cognitive benefits associated with physical exercise are widely acknowledged. However, exercise characteristics responsible for the improvement in cognition, along with the underlying mechanisms, remain to be elucidated. In an attempt to resolve these questions, our study compares cognitive outcomes after two distinct exercise modes in young adults. Specifically, we have examined the differences between high-intensity interval exercise (HIIE) and moderate-intensity continuous exercise (MIE) on visuospatial and declarative memory, contemplating the influence of exercise-associated parameters such as released lactate and non-modifiable factors, including BDNF polymorphism and biological sex. Our results demonstrate the relevance of exercise intensity and intensity-dependent lactate release in modulating cognitive improvement following exercise with no association with the non-modifiable factors mentioned above. These data suggest that the exercise characteristics and intensity of exercise sessions should be carefully considered to optimize cognitive response in young adults. Conversely, the consideration of genotype and biological sex is inappreciable.

**Abstract:**

Currently, high-intensity interval exercise (HIIE) is on the rise compared to moderate-intensity exercise (MIE) due to its similar benefits for health and performance with low time requirements. Recent studies show how physical exercise can also influence cognitive function, although the optimal dose and underlying mechanisms remain unknown. Therefore, in our study, we have compared the effects on visuospatial and declarative memory of different exercise intensities (HIIE vs. MIE), including possible implicated factors such as lactate released after each session and the Brain-Derived Neurotrophic Factor (BDNF) genotype. Thirty-six undergraduate students participated in this study. The HIIE session consisted of a 3 min warm-up, four 2 min sets at 90–95% of the maximal aerobic speed (MAS) with 2 min of passive recovery between sets, and a 3 min cooldown, and the MIE session implies the same total duration of continuous exercise at 60% of the MAS. Better improvements were found after HIIE than MIE on the backward condition of the visuospatial memory test (*p* = 0.014, η_p_^2^ = 0.17) and the 48 h retention of the declarative memory test (*p* = 0.04; d = 0.34). No differences were observed in the forward condition of the visuospatial memory test and the 7-day retention of the declarative memory test (*p* > 0.05). Moreover, non-modifiable parameters such as biological sex and BDNF polymorphism (Val/Val, Val/Met, or Met/Met) did not modulate the cognitive response to exercise. Curiously, the correlational analysis showed associations (*p* < 0.05) between changes in memory (visuospatial and declarative) and lactate release. In this sense, our results suggest an important role for intensity in improving cognitive function with exercise, regardless of genetic factors such as biological sex or BDNF Val66Met polymorphism.

## 1. Introduction

Evidence suggests that the practice of physical exercise has positive effects on cognitive function, with benefits attributed to both acute [1,2,3,4,5,6,7] and chronic exercise [2,5,6,7,8,9]. However, there is high variability in the observed effect sizes [10], which may be explained by the distinct characteristics of the physical exercise design, such as the type or dose, the cognitive domain assessed, or individual differences, such as the genotype [11]. Moreover, most studies on the relationship between exercise and cognitive function were conducted in children or older adults; meanwhile, adolescent and young adult populations remain underrepresented [7]. Therefore, although a large body of literature has focused on the study of the relationship between exercise and cognition, there is no clear agreement on the optimum dose of exercise and the associated mechanisms to produce greater improvements in cognitive function and even less so in young adults. The influence of common mediator variables, such as sex or genetic profile, on the aforementioned relationship also remains largely unresolved.

In an attempt to optimize current guidelines for exercise prescription to improve cognition, research has focused on the role of exercise dose in modulating the changes in cognitive function [1,3,9,12]. Insights gained from such research could explain the previously observed heterogeneity in response and reveal the underlying mechanisms. In their meta-analysis, Chang, Labban, Gapin, and Etnier [1] observed how, whenever there is sufficient rest after an exercise bout, very high-intensity (>93% Heart Rate) exercise would produce the greatest benefits in cognitive function. Along the same lines, a more recent review article [3] corroborated these observations concerning the intensity of the exercise, with high-intensity interval exercise (HIIE) accounting for greater improvements in inhibitory control, followed by vigorous-intensity and moderate-intensity continuous exercise (MIE). In agreement with these conclusions, it has been revealed that lower volume sessions (which, in turn, allow to sustain higher intensities) are associated with a better response, with sessions below twenty minutes, in particular, leading to superior effects [1,3]. Interestingly, following the dose–response perspective, recent studies have suggested that lactate released during high-intensity exercise may play an important role in acute cognitive enhancement in response to exercise [13]. Recent observations in mice have revealed how lactate released from exercising muscles during high-intensity bouts can cross the blood–brain barrier and induce BDNF expression in the hippocampus, improving learning and spatial memory [14]. Siebenmann et al. [15] proposed that the same mechanisms may hold true for humans. Lactate release has been previously correlated with cognitive performance in humans [13]. Moreover, as El Hayek, Khalifeh, Zibara, Abi Assaad, Emmanuel, Karnib, El-Ghandour, Nasrallah, Bilen, Ibrahim, Younes, Abou Haidar, Barmo, Jabre, Stephan, and Sleiman [14] have demonstrated, lactate enhanced BDNF production in the hippocampus, and BDNF, in turn, has been associated with hippocampus learning [16] and general memory processing [17].

Precisely, the arterio-jugular venous lactate difference was measured during incremental cycling exercise until exhaustion, concluding that brain lactate uptake during exercise was directly determined by the increase in arterial lactate concentration, with the latter rising in proportion to intensity. Moreover, lactate released during exercise could exert positive effects on brain health, modulating molecular pathways related to neurogenesis [18] and promoting neuronal survival [19].

The duration of the effects of exercise on cognition is another relevant topic in the literature [20,21]. In this sense, exercise could be linked to distinct phases of declarative memory. Thus, the effects of exercise have been evaluated, depending on the time point of application, in the acquisition (encoding and consolidation), storage, and retrieval stages [21]. It has been proposed that to isolate the effects of acute exercise on mechanisms related to memory consolidation, the exercise session should be performed after encoding, during the consolidation window, and not too close to the memory retrieval stage [21]. Addressing the memory consolidation phase has proven to be the most beneficial strategy for memory enhancement through exercise [20]. In this regard, exercise could transiently alter the availability of catecholamines (norepinephrine and dopamine) and neurotrophic factors such as the brain-derived neurotrophic factor (BDNF) [22] during consolidation, which could lead to an improvement in long-term memory.

Related to the secretion of BDNF, it is known that is regulated by the homonymous gene, which has a single nucleotide polymorphism (SNP) called SNP rs6262 or BDNF Val66Met gene polymorphism. This polymorphism substitutes a valine (Val) for methionine (Met) at codon 66. Individuals may present with either two valine SNPs (Val/Val), one valine and one methionine (Val/Met), or two methionines (Met/Met). The polymorphism exhibited by methionine (Met/Met) reduces BDNF secretion [23] and could explain the observed diminished response to exercise in terms of its effects on declarative memory in Met/Met carriers [24]. These findings allow us to infer the possible mediator role of the genomic profile and, in particular, the presence of BDNF Met/Met polymorphism on the effects of exercise on learning on the level of an individual.

In addition to the BDNF polymorphism, other aspects have been included in studies on exercise and cognition as possible moderators of the cognitive response. In this sense, biological sex is considered a powerful moderator [9,25,26]. BDNF release differs between sexes, but the impact and mechanisms of such differences remain unexplored [27]. At a physiological level, men should be able to release more BDNF after acute exercise [28]. Moreover, the BDNF genotype does not translate into cognitive benefits in females, but the opposite response has been reported in male subjects [29]. However, there is controversy when observing the cognitive response, with some studies reporting a better response to aerobic exercise in women [30,31] while others have reported worse outcomes in those studies with a higher percentage of female subjects [9]. Thus, ideally, studies should consider the inclusion of both sexes in their samples to increase the evidence in this regard.

In an effort to make up for the aforestated gaps in the literature and expand our knowledge of the mechanisms behind cognitive enhancement following physical exercise, we hypothesized the following: (i) HIIE would elicit better responses in short-term visuospatial memory and long-term formal memory consolidation than MIE, (ii) lactate release would be positively associated with cognitive improvements, (iii) subjects with the presence of at least one Met allele would score lower in memory tasks, and (iv) following exercise, the cognitive response in women would be amplified when compared to men.

## 2. Materials and Methods

### 2.1. Participants

Thirty-six undergraduate students (mean ± SD, age: 22.25 ± 2.94, height: 1.69 ± 0.09, weight: 70.03 ± 11.29, Body Mass Index (BMI): 24.25 ± 2.56, %fat mass: 22.73 ± 6.93, maximal aerobic speed (MAS): 14.82 ± 1.10, Maximum Heart Rate (HR_max)_: 196.22 ± 8.22) participated in this study. Prior to the study, all participants filled out the “Physical Activity Readiness Questionnaire” (PAR-Q) and signed written informed consent clarifying a range of measurements and exercise sessions contemplated within the study protocol and the anonymity and confidentiality of the extracted data. The experimental design was thought out according to the most recent revision of the Declaration of Helsinki and was approved by the Ethics Committee of the University (UMH.CID.DPC.02.17). The participants were advised to avoid strenuous physical exercise 24 h prior to every protocolled exercise session. In total, three sessions, separated by a week, were carried out, with each session scheduled for 9 a.m. Caffeine use before the exercise was discouraged. Water intake was restricted to 30 min prior to the visit to the laboratory, as indicated by the manufacturer of the saliva tests.

The sample size was calculated using G-power, where alpha was 0.05, and power was 0.8. We were looking for at least a medium effect size (0.01 < η_p_^2^ < 0.06). The dependent variable was TIME in the ANOVA RM TIME vs. Condition. With η_p_^2^ = 0.05 assumed for all the variables, the sample calculation for a priori ANOVA RM 2×2 was of 34 participants. In a post hoc analysis, the ANOVA RM showed a η_p_^2^ of 0.16 and 0.17, which means that the power of our study is 0.99.

### 2.2. Experimental Procedure

Participants attended the laboratory on three separate occasions. The first visit included the filling in of pre-participation questionnaires and informed consent by the participants, the collection of saliva samples to establish BDNF polymorphism, and a graded exercise test to determine the workload corresponding to the HIIE and MIE. The following two visits to the laboratory were randomized and counterbalanced to carry out the experimental conditions (HIIE or MIE) (Figure 1). For both visits, the participants were required to be present in the laboratory at 9 a.m.

The Visuospatial Memory was assessed before and after each experimental condition. The post-measurement was administered 15 min after the cessation of the exercise. Previous research has identified this time frame as optimal for maximizing the cognitive response to exercise [1,3]. The Long-Term Formal memory task was performed before each of the experimental conditions (the phase of encoding), thus placing the exercise session itself during the consolidation stage [21].

Blood lactate samples were drawn prior to exercise and 3 and 15 min upon cessation of the main exercise, coinciding with the end of the exercise protocol and marking the start of the post-measurements of the cognitive tasks.

The experimental protocol is summarized in Figure 2.

### 2.3. Experimental Conditions

The experimental conditions (HIIE and MIE) were carried out on a treadmill in randomized and counterbalanced order (Figure 1). The sessions were performed one week apart. Both sessions included a 3 min warm-up at 60% MAS. In the case of MIE, this intensity was sustained for the whole duration of the exercise (20 min). The HIIE session consisted of four bouts of 2 min at 90–95% MAS, interspersed with 2 min of passive recovery, and followed by a 3 min cooldown at walking speed to reach a total of 20 min. The volume of both sessions was, thus, standardized. The decision to limit the duration of both sessions to 20 min was founded in the previous research, where a trend towards better cognitive outcomes, in particular, in executive function, was observed in those sessions with similar volumes [1,3]. Heart rate (HR) was monitored using polar H7 chest straps and the Polar Beat app (Polar Electro Oy, Kempele, Finland) (Figure 3).

### 2.4. Measurements

#### 2.4.1. Graded Exercise Testing

A graded exercise test to volitional exhaustion was conducted on a treadmill to determine the MAS and the maximum HR. An HR measurement was monitored using the polar chest strap H7 (Polar Electro Oy, Kempele, Finland). After a 3 min warm-up at 5 km/h, the speed was increased by 1 km/h every minute until exhaustion. The treadmill was set at a gradient of 1% for both the warm-up and the main part. The participants were not allowed to drink or talk during the test and were asked to refrain from intense exercise for 24 h prior to the testing.

#### 2.4.2. Lactate Blood Sample

To determine the lactate concentrations, blood samples were obtained from the earlobes of the subjects at the beginning of the experimental session and 3 and 15 min after each of the main exercises in the experimental conditions using a portable lactate analyzer (Lactate Scout, SensLab GmbH, Leipzig, Germany).

#### 2.4.3. Genotype Analysis

Saliva samples were collected with an OrageneTM DNA Saliva Collection Kit (DNA Genotek S.L., Ottawa, ON, Canada). The DNA extraction protocol was provided by the manufacturer. The sample was further analyzed using a quantitative real-time StepOne PCR of the Applied Biosystem (Thermo Fisher Scientific S.A., Waltham, MA, USA), following the protocol of Sánchez-Romero et al. [32].

#### 2.4.4. Cognitive Function

##### Short-Term Visuospatial Memory Task

A digital version of the Corsi Block-Tapping task, widely used to assess visuospatial and short-term working memory [33] was used. The protocol described by Kessels, van Zandvoort, Postma, Kappelle, and de Haan [33] contemplates two modalities (Forward and Backward) within the same task. Nine items are visible on the screen, and for each trial, these items are “lit” one by one, at a rate of one item per second, in a randomized order. The first modality (Forward) requires that the participants tap on the items in the same order as they first appeared on the screen, immediately after the last item of the sequence has been displayed. For the second modality (Backward), on the other hand, the participants are instructed to repeat the sequence in reverse order. The participants receive two trials for every sequence with the same number of items. If at least one of the attempts is successful, the next sequence of a greater length is administered. The test is over once the participant fails to replicate two consecutive sequences of the same length. Given the test characteristics, two scoring alternatives can be obtained: on the one hand, the number of successful trials achieved until the end of the test, and on the other hand, the block span (which is equal to the length of the last sequence) for each of the test modalities. The product of both scores gives rise to the total score, which has been shown to be the most reliable variable for assessing changes in this visuospatial memory test [33]. Therefore, the test evaluates the following four variables: (i) block span forward (CSF), (ii) total score forward (TSF), (iii) block span backward (CSB), and (iv) total score backward (TSB). In this sense, at the baseline, we observed scores very similar to those reported by Kessels, van Zandvoort, Postma, Kappelle, and de Haan [33] in healthy individuals. On the other hand, the scores were also very similar at the pre-test of both experimental conditions for both the forward (*p* = 0.75) and backward (*p* = 0.91) modalities.

##### Long-Term Formal Memory Task

For the formal memory task, the subjects were required to memorize a 15-line text (221 words) with factual information [34] about the 1954 Soccer World Cup or the 1967 Handball World Championship. The participants were given 10 min before each experimental situation (MIE and HIIE sessions), thus placing the latter after memory encoding. Memory recall, in turn, was tested following [21] 48 h and seven days using a list of questions on ten factual items from either of the two texts [34]. For each correct answer, the participants were awarded one point. Each one of the two texts was counterbalanced with the experimental condition to eliminate a possible contaminating effect of the text.

### 2.5. Statistical Analysis

The alpha was set to 0.05 for all analyses. The normality of the data set was examined through the Shapiro–Wilk test. Before the main analysis, paired *t*-tests were performed to compare the exercise intensity variables between the MIE and HIIE sessions and to examine possible differences in the pre-assessment of visuospatial memory in both experimental conditions.

To analyze the differences in the visuospatial memory between the experimental conditions, a repeated-measures analysis of variance (ANOVA RM) was performed with two within-subject factors (Time × Condition). Sphericity was computed with Mauchly’s sphericity test.

To analyze the differences in long-term memory (recall) on the formal memory test and between the MIE and HIIE intensities, paired *t*-tests or the Wilcoxon signed rank were also calculated according to the data distribution.

Two additional ANOVA RM analyses were conducted separately for BDNF polymorphism (Val/Val, Val/Met, or Met/Met) or sex (male or female) as a between-subject factor to analyze the possible influences of these non-modifiable factors. In the case of finding statistically significant differences in any of the ANOVA RM analyses, post hoc analyses with Bonferroni adjustment were employed.

As a measure of the effect sizes on the ANOVA RM analysis, partial eta-squared (η_p_^2^) was used. The effect sizes are expressed as partial eta-squared (η_p_^2^) and are grouped as small (≤0.01), medium (≤0.06), and large (≤0.14) [35]. Finally, Pearson’s correlation analyses (r) were used to establish possible associations between the changes (Δ) in blood lactate and cognitive function [13].

All the results were analyzed using the JASP 0.16 software (Eric-Jan Wagenmakers, Department of the Psychological Methods University of Amsterdam, Nieuwe Achtergracht 129B, Amsterdam, The Netherlands).

## 3. Results

### 3.1. Exercise Characteristics

Both, absolute (164.1 vs. 152.6 bpm, *p* < 0.01) and relative (83.6 vs. 77.8%, *p* < 0.01) HR values were higher in HIIE than MIE for the whole 20 min duration of the session. The average for the HIIE intervals was 181.5 bpm, which corresponded to 92.5% of the HR_max_, and the average during that same length of time in the MIE session was 155.7 bpm (79% HR_max_). Regarding the lactate concentrations, significant differences between HIIE and MIE were found at both the 3- (10.18 vs. 1.86 mmol, *p* < 0.01) and 15- (8.17 vs. 1.48 mmol, *p* < 0.01) minutes post-exercise measurements.

### 3.2. Intensity Impact on Memory

Visuospatial memory: The repeated-measures ANOVA revealed the absence of the significant interaction of Time x Condition for CSF [*F*_(1,35)_ = 0.97, *p* = 0.33, η_p_^2^ = 0.03] or TSF [*F*_(1,35)_ = 0.49, *p* = 0.49, η_p_^2^ = 0.014]. In contrast, there were significant differences for CSB [*F*_(1,34)_ = 6.67, *p* = 0.014, η_p_^2^ = 0.16] and TSB [*F*_(1,34)_ = 6.79, *p* = 0.014, η_p_^2^ = 0.17]. The post hoc Bonferroni analysis for TSB showed significant differences in pre vs. post HIIE [t(34) = 2.87, *p* = 0.03] and post HIIE vs. post MIE [t(34) = 2.95, *p* = 0.03] in favor of the HIIE session. In CSB, there was a significant difference in post HIIE vs. post MIE [t(34) = 2.97, *p* = 0.02] in favor of the HIIE session (Figure 4).

Formal/Declarative memory test: Paired *t*-tests showed significant differences in the recall of factual information at 48 h in favor of the HIIE session (*p* = 0.04; d = 0.34), which was not the case with recall at 7 days (*p* = 0.79; d = 0.057) (Figure 5).

### 3.3. Interaction of Response with Sex and BDNF Polymorphism

To analyze possible differences in the cognitive response dependent on non-modifiable factors, sex and Val66Met were added to the analyses. From the genetic analysis, we found 10 subjects with the Val/Val coding, 15 with the Val/Met coding, and 11 with the Met/Met coding.

No significant differences were found in the Time x Condition x Sex interaction in CSF [*F*_(1,34)_ = 1.93, *p* = 0.18, η_p_^2^ = 0.054], CSB [*F*_(1,33)_ = 0.09, *p* = 0.77, η_p_^2^ = 0.003], TSF [*F*_(1,34)_ = 2.75, *p* = 0.11, η_p_^2^ = 0.07], nor TSB [*F*_(1,33)_ = 0.10, *p* = 0.75, η_p_^2^ = 0.003]. No significant differences were found in the Condition x Sex interaction in the FM 48 h recall [F_(1,34)_ = 1.59, *p* = 0.22; η_p_^2^ = 0.04] nor FM 7 days recall [F_(1,34)_ = 0.001, *p* = 0.97; η_p_^2^ = 0.006]. No between-subject effects were found for any of the included variables based on the sex factor (*p* > 0.05).

No significant differences were found in the Time × Condition × BDNF Val66Met interaction in CSF [*F*_(2,33)_ = 0.69, *p* = 0.51, η_p_^2^ = 0.04], CSB [*F*_(2,32)_ = 0.38, *p* = 0.69, η_p_^2^ = 0.02], TSF [*F*_(2,33)_ = 1.55, *p* = 0.23, η_p_^2^ = 0.08], nor TSB [*F*_(2,32)_ = 0.48, *p* = 0.62, η_p_^2^ = 0.03]. No significant differences were found in the Condition × BDNF Val66Met in any formal memory recall (*p* < 0.05). No between-subject effects were found for any of the included variables based on the BDNF Val66Met gene factor (*p* > 0.05).

### 3.4. Correlation Analysis

Positive correlations were identified between the changes (Δ) in the LA concentration at 3 min (CSB, r = 0.295, *p* = 0.013; TSB, r = 0.363, *p* = 0.002; and Recall 48 h, r = 0.225, *p* = 0.058) and at 15 min (CSB, r = 0.324, *p* = 0.006; TSB, r = 0.428, *p* < 0.001; and Recall 48 h, r = 0.264, *p* = 0.025) (Figure 6).

## 4. Discussion

Our results indicate an acute performance enhancement in the backward modality of the short-term visuospatial memory task. In this sense, some studies have highlighted the higher complexity of the backward task [36,37]. This could be explained by the participation of the distinct working memory subsystems in the two modalities, where the backward condition puts a heavier load on the central executive (e.g., to reverse the sequence) [36,37]. Previous studies exploring the relationship between exercise and memory have generally observed that exercise could have little [38] or null [39] impact on the said cognitive domain. However, more recently, the positive effects of acute exercise on both short- and long-term memory were shown [10]. In this line, Chang, Labban, Gapin, and Etnier [1] have evaluated the effects of exercise on different types of memory and observed improvements only in free recall and short-term visual memory. In their meta-analysis, they also observed that as long as there is a rest period after the exercise session, the exercise of greater intensity would lead to superior gains in cognitive function, something that we have been able to corroborate with our results. Along the same lines and consistent with our findings, a more recent meta-analysis revealed that the superiority of HIIE concerned executive function, while moderate exercise had only a small effect [3]. Therefore, optimizing exercise protocols could be a key element in reducing variability in the cognitive response. As such, current research proposes the exploration of different types or doses of exercise to improve the cognitive response in young adults, with findings similar to ours, where the benefits of HIIE for the executive function are superior compared to MIE [40,41,42].

On the other hand, in addition to the acute effects observed in the literature on short-term visuospatial memory [1,10], it has been suggested that acute physical exercise can benefit long-term declarative memory [10]. In this sense, we have observed a significant positive effect of exercise intensity on memory recall at 48 h, in favor of the HIIE session, in the long-term formal memory task [34]. Furthermore, we have placed the exercise session before the memory encoding stage with the intention of testing the impact of exercise intensity on mechanisms related to memory consolidation [21]. This particular phase of memory processing has been reported to be affected by exercise through the production of neurotrophins and growth factors [21]. Thus, it has been revealed how exercise plays a central role in altering mediators such as BDNF and hippocampal cell proliferation, with the latter two affecting the exercise–memory relationship [43]. Our results suggest that the HIIE session during the consolidation phase favors information recall at 48 h compared to the MIE session. In this sense, and relating it to the mechanisms mentioned above, Saucedo Marquez et al. [44] observed that HIIE was a more powerful stimulus to increase the systemic release of BDNF compared to MIE. The same study also established that a greater synthesis of BDNF in the brain could explain the increased systemic levels of this neurotrophic factor and account for the observed favorable effects of HIIE versus MIE in our study.

Related to the mechanisms underlying the positive effect of HIIE over MIE on memory mentioned above, it is worth clarifying that HIIE must be differentiated from high-intensity training, with the latter resulting in impaired cognitive performance [45,46]. The main advantage of interval exercise (HIIE) is the abundant lactate production but limited catecholamine release and mental stress [42]. Interestingly, we have found an association between blood lactate concentrations and improvements in short-term visuospatial and long-term declarative memory. This observation has been recently brought to attention, revealing the critical role of lactate in cognitive enhancement and brain health after HIIE [13]. It has been previously demonstrated that lactate can cross the blood–brain barrier, and once attached to a specific receptor, it triggers multiple reactions in the brain [14,47]. Furthermore, Skriver, Roig, Lundbye-Jensen, Pingel, Helge, Kiens, and Nielsen [22] have found a correlation between systemic lactate release and the acquisition and retention of motor skills. Regarding its effects on metabolism, during HIIE, the brain becomes increasingly dependent on lactate supply [48] in contrast to preferential glucose uptake in resting conditions. Cerebral lactate consumption rises once the blood lactate levels are above ≥2 mM [49], which the values far exceeded in our HIIE session but not in MIE. As we have highlighted above, one of the hypotheses of the cognitive improvement of HIIE versus MIE implies the mediator role of BDNF, which could be linked to increased lactate production during exercise. Along the same lines, El Hayek, Khalifeh, Zibara, Abi Assaad, Emmanuel, Karnib, El-Ghandour, Nasrallah, Bilen, Ibrahim, Younes, Abou Haidar, Barmo, Jabre, Stephan, and Sleiman [14] have suggested that lactate released during exercise is transported to the brain, crossing the blood–brain barrier, where it induces BDNF expression. Curiously, this exercise-induced lactate shuttle to the brain has been shown to enhance visuospatial memory [14]. Other effects of lactate on the central nervous system include the support of synaptic activity [50], long-term potentiation and memory formation [51], and neuronal plasticity [52], all of which suggest that brain function, understood as cognitive performance, depends on the lactate metabolism in the brain. Taken together, these observations could be behind the associations we found between memory enhancement and higher lactate concentrations after the HIIE session.

When proposing the hypothesis that relates lactate and BDNF, it is worth considering the BDNF polymorphism. We have found a similar relative distribution between Val/Val and Met carriers (Val/Met or Met/Met) in our study to those previously reported [53]. Given that the presence of Val66Met could result in decreased levels of BDNF released in response to exercise [54], this SNP should be considered when analyzing the effects of exercise on cognition. In human subjects, some studies have reported an interplay between physical exercise and declarative memory influenced by the BDNF polymorphism, where a positive influence was only revealed in the Val/Val carriers [24]. However, divergent theories elucidating the effects of distinct SNPs on the exercise-cognition relationship exist. For instance, and in agreement with the findings above, the presence of valine in a gene sequence could predispose one to enhanced cognitive response following exercise, and this effect could be explained by the increased release of BDNF [55]. However, contrasting hypotheses favoring the opposite conclusion exist. More precisely, it has been suggested that Met carriers could experience exercise-induced cognitive gains to a larger extent, where exercise could make up for lower baseline levels of released BDNF [56]. In our study, we could not find any association between the BDNF polymorphism with the cognitive response following exercise in young adults. Unfortunately, this observation does nothing but add up to the conflicting evidence on the role of BDNF polymorphism in the exercise–cognition relationship [11]. Along this line of thought, and in parallel with the findings by de Las Heras, Rodrigues, Cristini, Weiss, Prats-Puig, and Roig [11], this disparity in the reported results could indicate that, in fact, the cognitive gains associated with physical exercise are not dependent on the genetic profile, and all individuals could derive benefits from exercise on cognitive function.

Finally, we have attempted to determine if the differences in the cognitive response observed after HIIE and MIE depended on sex. In this regard, we have not found any discrepancies, and the trend of HIIE being the superior modality was true for both sexes. There are some indications that men release more BDNF both after acute exercise [28] and after a period of training [28], which could be translated into improved cognitive function. However, there is no agreement in the literature with regard to functional changes, with some studies observing a better cognitive response to aerobic exercise in women [30,31], while others have found worse long-term outcomes in cognitive function after exercise programs in those studies with a higher percentage of female participants [9]. This inconsistency could be explained by methodological differences between studies, including the preferential selection of older adults or major variations in the dose/intensity prescription of the chosen exercise type [9]. Therefore, although we have not found any sex-dependent differences in our case, future research on exercise and cognition should contemplate equivalent samples of both sexes to understand the possible variations between them better.

This study has some limitations that must be considered when interpreting the results. First, we inferred the association between accelerated brain lactate metabolism and improved cognitive function from previous research [40]. However, studies directly measuring brain lactate levels are required to test our hypothesis about the relationship between lactate concentrations and cognitive function. Second, given the large body of evidence on the effects of physical exercise on memory [10], our objective was to compare different intensities considering the role of possible moderators, such as lactate and BDNF polymorphism, in this relationship. However, just as in the study by Tsukamoto, Suga, Takenaka, Tanaka, Takeuchi, Hamaoka, Isaka, and Hashimoto [40], we have not included a sedentary control condition; thus, we could not assess the ecological context where participation in physical exercise would have contrasted with a different condition. These alternative situations, in turn, could influence the cognitive response [57].

Moreover, the training load (a product of the intensity and duration of the exercise session) varies between experimental protocols, and we are aware that the magnitude of the training load will determine the physiological adaptations following an exercise bout. The reason for limiting both protocols to 20 min is founded in the previous research, where this timeframe has provided the most benefit for cognitive function [3,58,59]. Thus, once we have settled upon the optimal duration, we can only manipulate the exercise intensity, which would inevitably produce different loads and could be considered a limitation of the present study. However, the issue is the lack of previous evidence on the relationship between exercise load and cognitive function.

In future research, the focus should be placed on the examination of the dose–response relationship as it concerns the intensity-cognitive response link, but it is also fundamental to explore the remaining FITT (frequency, intensity, time, type) variables and the relationship between them (exercise load). Many questions still remain unanswered regarding the impact of the FITT dimensions, and, of course, we are still far from fully elucidating the implications of load in exercise-cognition research.

However, we also view our work as an important step towards increasing our comprehension of the exercise–cognition relationship in young adults. First, we have examined this relationship from a dose–response perspective, an aspect that has received less attention as a modifiable factor [12]. To augment our understanding of the possible mechanisms underlying the effects of exercise on cognition, we have also included the analysis of lactate and BDNF polymorphism. Lactate has been related to exercise intensity and BDNF polymorphism is a non-modifiable genetic condition; both were previously listed as possible modulators in the cognitive response to exercise, with, nonetheless, a lack of consideration in the relevant literature so far [7]. Finally, research to date has mostly failed to include all age groups. As such, children and older adults are preferentially selected as study populations. Meanwhile, adolescents and young adults have been the objective of far fewer studies [7]. Therefore, the inclusion of young people in our work contributes to the lacking evidence supporting the role of physical exercise in enhancing cognitive function in this age group and proposing the possible mechanisms underlying the exercise–cognition relationship [7].

## 5. Conclusions

Our main findings suggest that HIIE is superior for enhancing memory compared to MIE. These positive effects, observed after HIIE but not after MIE, confer an important role to exercise intensity in magnifying cognitive response after acute physical exercise in young adults. Moreover, we have observed a positive association between blood lactate concentrations and memory improvements, suggesting that lactate released from the exercising muscle during HIIE may underlie the differences in cognitive function after two modalities. These observations apply to improvements in both short-term visuospatial memory, immediately after an exercise session, and long-term declarative memory. In the latter case, a session of HIIE during the memory consolidation phase resulted in enhanced formal memory retention and recall after 48 h. The number of recollected items after HIIE was far superior compared to MIE; furthermore, their total count correlated with the blood lactate concentrations after acute exercise. Finally, non-modifiable parameters such as biological sex and BDNF genetic polymorphism did not modulate the cognitive response to exercise.

## Figures and Tables

**Figure 1 biology-11-01541-f001:**
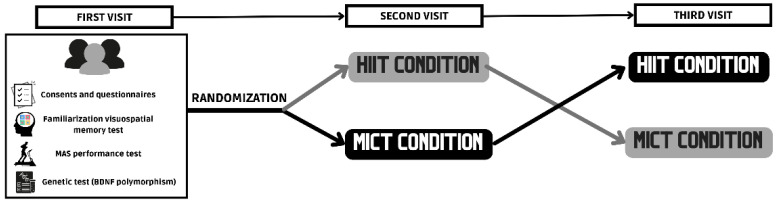
Study design.

**Figure 2 biology-11-01541-f002:**
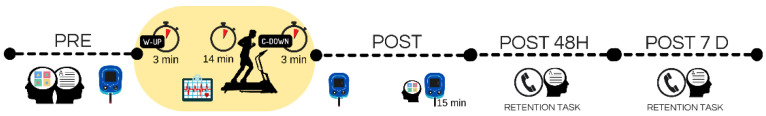
Experimental protocol design.

**Figure 3 biology-11-01541-f003:**
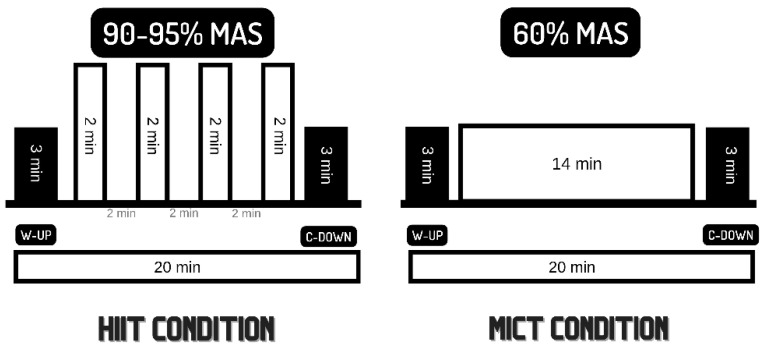
Exercise session design.

**Figure 4 biology-11-01541-f004:**
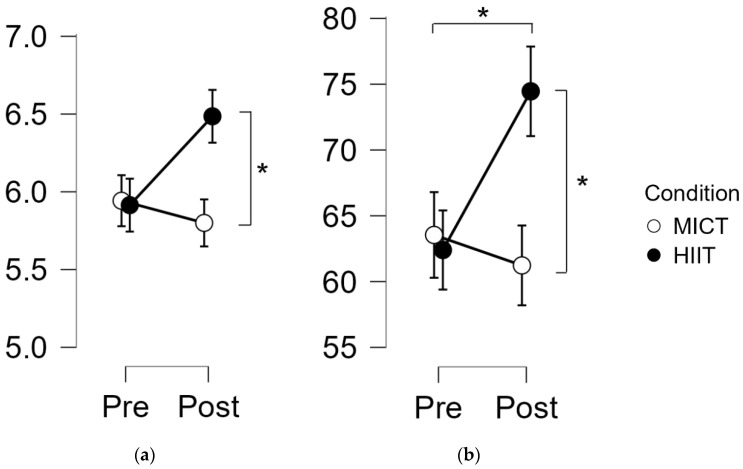
Visuospatial memory results. (**a**) Corsi Span Backward; (**b**) Total Score Backward. *****: Indicates a significant effect of time or between conditions (*p* < 0.05).

**Figure 5 biology-11-01541-f005:**
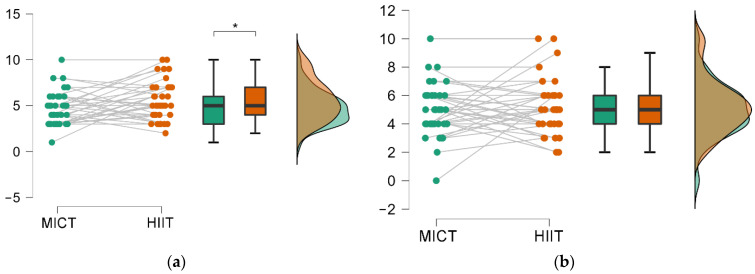
Raincloud plots for the formal memory results. (**a**) Recall at 48 h; (**b**) Recall at 7 days. *: Indicates a significant effect of time or between conditions.

**Figure 6 biology-11-01541-f006:**
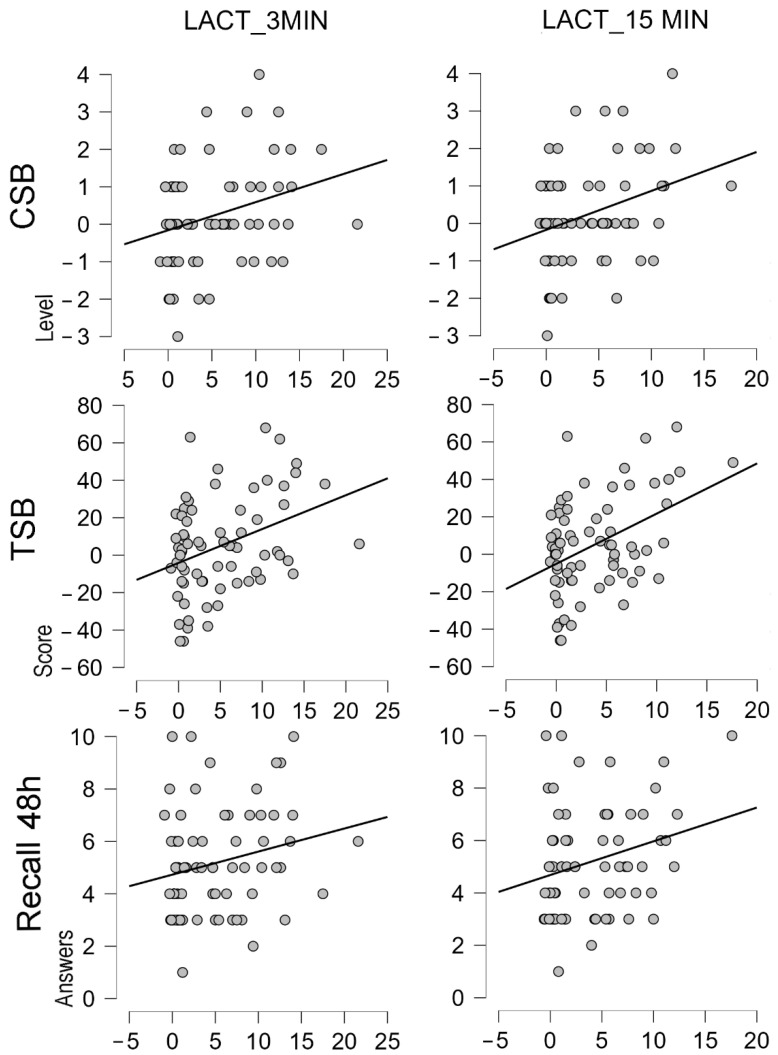
Correlations between changes (Δ) in LA concentration and Δ memory scores. CSB: Corsi Span Backward, TSB: Total Score Backward, Recall 24 h: Formal memory recall at 48 h.

## Data Availability

Not applicable.

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
