# Peer review of "Memory Modulation by Exercise in Young Adults Is Related to Lactate and Not Affected by Sex or BDNF Polymorphism"

_biology, 2022, doi:10.3390/biology11101541_

Round 1

Reviewer 1 Report

First of all I want to thank the authors for choosing this journal, as well as the effort in carrying out the research.

Unfortunately, the work presents great limitations both in its design and in obtaining the results, for this reason, and much to my regret, I must classify it as rejected.

In the first place, the authors indicate in the methods that they have only left 24 hours without performing any type of vigorous activity, when it is widely known that defects of high neuromuscular or metabolic demand can generate fatigue that can affect the organism up to 72 hours later. One of the affected mechanisms is the activity of glycolytic enzymes, severely affecting the production of lactic acid. This fact has been able to greatly influence the results and conclusions obtained from the investigation.

Second, in section 2.3. of the methods, the authors state that the warm-up consisted of a 3-minute activity at an intensity of 60% of the MAS. This warm-up is inappropriate for performing glycolytic work, as it is insufficient to sufficiently activate the aerobic and anaerobic metabolic pathways. In addition, the intensity of the warm-up is considerably lower than the anaerobic threshold, this being an essential intensity to prepare the body for activities of a glycolytic nature.ç

Lastly, and what in my opinion is the greatest limitation of the work is that the two protocols used have a completely different load. As both sessions (HIIT and MICT) have the same duration (20 minutes) but very different intensities, the total value of the load and, therefore, of the generated training stimulus, are very different.

This is demonstrated by the results indicated in line 262, where much higher values ​​are obtained in lactate production and in heart rates in HIIT sessions.

For this reason, the comparison between stimuli is erroneous, since two very different stimuli are being compared in terms of their magnitude. The correct thing in this situation should have been to equate the load of both sessions, modifying the duration, to obtain a similar stimulus (in terms of energy expended, VO2 or other physiological criteria) in terms of magnitude.

Additionally, the blood lactate measurements have not been accompanied by an evaluation of the hematocrit, so they are not reliable since the state or the state of hydration or the hemoconcentration values ​​are not known. It is well known that the latter can be modified by exercise-induced sweating and markedly modulate blood lactate concentrations.

For all these reasons, I am sorry to classify this work as rejected, and I would like to encourage the authors to have this journal again on future occasions.

Author Response

Answer to Reviewer 1

First of all I want to thank the authors for choosing this journal, as well as the effort in carrying out the research.

Unfortunately, the work presents great limitations both in its design and in obtaining the results, for this reason, and much to my regret, I must classify it as rejected.

  • First of all, we appreciate the time you´ve put into your comments. It´s a shame that you have made such a decision regarding our work; nonetheless, we are happy to address all your concerns and, hopefully, change your point of view.

In the first place, the authors indicate in the methods that they have only left 24 hours without performing any type of vigorous activity, when it is widely known that defects of high neuromuscular or metabolic demand can generate fatigue that can affect the organism up to 72 hours later. One of the affected mechanisms is the activity of glycolytic enzymes, severely affecting the production of lactic acid. This fact has been able to greatly influence the results and conclusions obtained from the investigation.

  • The reviewer, of course, is right, and 72 hours of recovery are necessary to ensure optimal performance. However, our participants are young college students who engage in recreational physical activity. It is highly improbable that they would have accumulated as much metabolic and neuromuscular strain as to require these long-lasting recovery periods. Moreover, we could not find any publication that could verify a marked reduction in lactate following incomplete recovery. However, it should be noted that the purpose of our research was to correlate lactate production with cognitive enhancement. As demonstrated in our study, low levels of lactate released were indeed associated with less improvement in cognitive tasks. Thus, consistent with our hypothesis, there is a link between lactate levels and cognitive response. Our study aimed to investigate the acute effects of exercise and lactate release on cognitive performance. Conversely, impaired physical performance, as a consequence of incomplete recovery, would not have necessarily modified the cognitive response following an exercise bout. Moreover, our participants are far from being classified as elite collegiate athletes (the mean MAS value is only 14 km/h). It seems unlikely that at any point of the research process, they have accumulated extended periods of highly demanding training, and thus, that the latter could affect their affective or cognitive state to an important degree.

Second, in section 2.3. of the methods, the authors state that the warm-up consisted of a 3-minute activity at an intensity of 60% of the MAS. This warm-up is inappropriate for performing glycolytic work, as it is insufficient to sufficiently activate the aerobic and anaerobic metabolic pathways. In addition, the intensity of the warm-up is considerably lower than the anaerobic threshold, this being an essential intensity to prepare the body for activities of a glycolytic nature.

  • Once again, the recommendations given by the reviewer, in this case concerning the warm-up, are accurate. However, metabolic inefficiency as the result of insufficient warm-up, it did not necessarily confound our results. A warm-up of longer duration would have amplified aerobic response, and as a result, it is plausible that less lactate would have been produced. Conversely, an accentuated anaerobic metabolism from the onset of exercise would imply an increase in lactate production. As we have demonstrated, higher lactate levels also correlated with acute cognitive enhancement in a manner dependent on the intensity of an exercise bout, in accord with individual lactate release.

Lastly, and what in my opinion is the greatest limitation of the work is that the two protocols used have a completely different load. As both sessions (HIIT and MICT) have the same duration (20 minutes) but very different intensities, the total value of the load and, therefore, of the generated training stimulus, are very different.

  • The observations made by the reviewer concerning the load of two training protocols are correct. However, there are a few reasons why rather than designing protocols of equivalent loads, the exercise-cognitive research field has focused on the session duration to optimize the cognitive response. In line with this view, Oberste et al. (2019) have concluded in their meta-analysis:

“When the studies were clustered in duration subgroups (duration up to 20 min vs. duration between 21 and 40 min vs. more than 40 min), subgroups with shorter exercise duration apparently yielded more beneficial effects (−0.32 vs. −0.18 vs. −0.07). If all available information (absolute value of exercise duration) was included in the analysis (meta-regression), the positive relationship between exercise duration and magnitude of effects became even clearer.”

Oberste, M., Javelle, F., Sharma, S., Joisten, N., Walzik, D., Bloch, W., & Zimmer, P. (2019). Effects and moderators of acute aerobic exercise on subsequent interference control: a systematic review and meta-analysis. Frontiers in psychology, 10, 2616.)

We are well aware that distinct exercise loads would translate into different physiological adaptations and fatigue patterns, but this is not as applicable to exercise and cognition research. The latter has traditionally explored the relationship between cognitive response and exercise intensity through the lens of optimal session duration, as has been previously used in a number of relevant studies (Chang et al., 2015; Chang et al., 2019; Shi et al. 2022).

This is demonstrated by the results indicated in line 262, where much higher values ​​are obtained in lactate production and in heart rates in HIIT sessions.-

  • The aim of the study was to identify the implications of manipulating the exercise intensity for cognitive improvement and the molecular mechanisms linking the observed cognitive enhancement to lactate release. Hence, the requirement to design two almost contrasting exercise protocols of different intensities.

For this reason, the comparison between stimuli is erroneous, since two very different stimuli are being compared in terms of their magnitude. The correct thing in this situation should have been to equate the load of both sessions, modifying the duration, to obtain a similar stimulus (in terms of energy expended, VO2 or other physiological criteria) in terms of magnitude.

  • As we mentioned in the paragraphs above, our intention was never to evaluate physiological adaptations to exercise training; but to examine the dose-response relationship between exercise intensity and acute cognitive response. A 20-minute session has been recognized as an optimum duration to observe changes in cognitive performance. Thus, if two protocols have different intensities, the load would necessarily be modified if the duration remains constant. It is also worth noting that exercise-cognition research is still exploring the optimum combination of the FITT principle dimensions (frequency, intensity, time, type), far away from the concept of load.

The comparison of different loads in and of itself is problematic, given the low evidence we have so far on the FITT variables. Moreover, we can not assume that load is more important for acute cognitive response than the FITT variables alone, as the evidence on the impact of load on cognitive response is lacking.

Additionally, the blood lactate measurements have not been accompanied by an evaluation of the hematocrit, so they are not reliable since the state or the state of hydration or the hemoconcentration values ​​are not known. It is well known that the latter can be modified by exercise-induced sweating and markedly modulate blood lactate concentrations.

  • The reviewer´s observations on the relevance of hydration are well noted. It is indeed of great concern if severe dehydration occurs in participants. However, our protocol was only 20 minutes long including warm-up and cooldown. Both experimental conditions also took place in a 20-degree Celsius facility, and thus, it seems unlikely that dehydration processes could have modified lactate response to a relevant degree. Moreover, our exercise session is similar in duration to a standard graded exercise test, where lactate is typically measured to determine the lactate threshold. The procedure of such tests also requires even higher intensities; yet, to the best of our knowledge, hematocrit is rarely evaluated in these protocols (Pettitt et al., 2013).

Pettitt, R. W., Clark, I. E., Ebner, S. M., Sedgeman, D. T., & Murray, S. R. (2013). Gas exchange threshold and vo2max testing for athletes: an update. The Journal of Strength & Conditioning Research27(2), 549-555.

For all these reasons, I am sorry to classify this work as rejected, and I would like to encourage the authors to have this journal again on future occasions

  • We are hoping our response can change the reviewer´s decision.

Reviewer 2 Report

I would like to first commend the authors on a well-developed project and an excellent presentation of their study and outcomes.  The need for work on exercise dose response is of the utmost priority in the exercise sciences and preventative and therapeutic medicine.  I strongly encourage the authors continue this path of work in the future.  I have a few considerations for the article to improve the already exceptional readability for the intended audience.

Abstract - Include the prescription used for MICT

Line 55 - Eliminate “even less so in younger subjects” as you previously state that most work is done in children, this is contradictory - unless you change subjects to adults here?

Line 63 - Eliminate the “the” preceding “high-intensity exercise would…”

Section 2.3, Lines 165-176.  Figure 3 doesn’t clearly match the description contained here.  Ex. The MICT is described as 3-min at 60% MAS for warm-up following the remainder of 20-min. at that same intensity but in Figure 3, it only shows a 14 min exercise period.  It may need clarification that the warm-up and cool-down times are included in the 20 min.?  Consider adding the 3-min warm up and 3 min cool down to the figure or change the text wording to indicate the MICT is 14 minutes long on line168.

Section 3.1, Line 264. Please include the mean HR for MICT as well.

Figure 5- Consider changing the fill texture or adding more labels as when the figure is converted to black and white for printing it is hard to differentiate.

Section 3.3- It is unclear if the last statement of the second and third paragraph includes no differences in exercise intensity between the variables?  If not - can the authors mention if there are intensity differences between genders or Val66Met?

Figure 6 - Please add labels to the y-axis.

Author Response

Answer to Reviewer 2

I would like to first commend the authors on a well-developed project and an excellent presentation of their study and outcomes.  The need for work on exercise dose response is of the utmost priority in the exercise sciences and preventative and therapeutic medicine.  I strongly encourage the authors continue this path of work in the future.  I have a few considerations for the article to improve the already exceptional readability for the intended audience.

  • We are grateful for the time the reviewer took to critique our manuscript. We are hoping to resolve all his concerns.

Abstract - Include the prescription used for MICT

  • In line 31-33 we have included the following: “and MICT session implies the same total duration of continuous exercise at 60% of MAS.”

Line 55 - Eliminate “even less so in younger subjects” as you previously state that most work is done in children, this is contradictory - unless you change subjects to adults here?

  • The sentence in line 57 has been modified: “even less so in young adults”, as we want to highlight the low amount of literature available on this population.

Line 63 - Eliminate the “the” preceding “high-intensity exercise would…”

  • “The” has been erased in line 65.

Section 2.3, Lines 165-176.  Figure 3 doesn’t clearly match the description contained here.  Ex. The MICT is described as 3-min at 60% MAS for warm-up following the remainder of 20-min. at that same intensity but in Figure 3, it only shows a 14 min exercise period.  It may need clarification that the warm-up and cool-down times are included in the 20 min.?  Consider adding the 3-min warm up and 3 min cool down to the figure or change the text wording to indicate the MICT is 14 minutes long on line168.

  • Thank you for this commentary, an updated version of figure 3 has been included:

         You can see the image in the PDF

Section 3.1, Line 264. Please include the mean HR for MICT as well.

  • Thank you for this observation, the mean HR for the central 14 minutes of MICT is now included in line 283: “The average for the HIIE intervals was 181.5 bpm, which corresponded to 92.5% of HRmax, and the average during that same length of time in the MIE session was 155.7 bpm (79% HRmax)”.

Figure 5- Consider changing the fill texture or adding more labels as when the figure is converted to black and white for printing it is hard to differentiate.

  • The Raincloud plots from JASP are necessarily in color. We can introduce a simpler black-and-white graph, but it will display less information; in fact, only mean values could be included in such a graph.

Section 3.3- It is unclear if the last statement of the second and third paragraph includes no differences in exercise intensity between the variables  If not - can the authors mention if there are intensity differences between genders or Val66Met?

  • The statement indicates that no differences were observed between sexes, nor between BDNF SNP and any other variables. Intensity is included among these variables. No differences between sex and intensity or lactate were revealed in T-Test, neither between BDNF polymorphisms and intensity or lactate measured with ANOVA.

Figure 6 - Please add labels to the y-axis.

  • Labels have been included in the Y axis of the figure 6.

Reviewer 3 Report

The manuscript “Memory modulation by exercise in young adults is related to lactate and not affect by BDNF polymorphism” aimed to investigate the impacts of different-intensity exercises, BDNF polymorphism and sex on memory modulation. I commend the authors for their work. General comments and specific points and sections are provided below:

General comments

Presentation: the presentation is good and the writing is sound, with no need for professional revision. I recommend that the authors screen the manuscript for minor typos. Some of the abbreviatures used in the study are not defined (e.g., MAS, FCmax, BMI, etc).

Novelty: the topic itself is not novel, but the perspective adopted in the study is.

Title: the title is well written, but does not cite the sex aspect of the study. Please consult other comments on the use of the terms MICT and HIIT.

Abstract: overall, the abstract is well written and provides the essential information. I did miss the description of MICT sessions in the abstract. Also, the opening statement (“HIIT is on the rise compared to MICT due to its benefits for health and performance”) can be misleading as a possible superiority of HIIT over MICT is still under heated debate in the literature.

Introduction: the introduction is well written and briefly presents some of the topics the study aims to investigate. However, probably due to the many aims and hypotheses presented, the rationale is relatively shallow, with as little as a couple of lines invested in presenting sex-related differences in BDNF secretion and cognition. The link between lactate and cognition/memory modulation should also be further addressed, as it appears to be the main argument of the study.

Materials and Method: methods are well described and adequate. I do have one important point to address regarding the choice of the HIIT vs MICT protocols. This comment is presented in the discussion topic below.

Results: the results are clearly presented and I commend the authors for their work in presenting such a large amount of data and yet managing to be clear and concise.

Discussion: the discussion is comprehensive and thorough. All important aspects of the study are clearly addressed and contextualized with the current literature. I have one major concern which mostly related to terminology. The authors choose to adopt the terms HIIT and MICT, but the characteristics of the MICT session are not in line with actual moderate intensity continuous training guidelines. MICT should be performed with longer durations to elicit its benefits. The HIIT session appears to be adequate in the study. I suggest that the authors change the terminology in the manuscript to make it clear that what is being compared in the study is the intensity of the exercise (severe vs moderate), rather than the modality (HIIT vs MICT). I would expect that a comparison between HIIT and MICT would be controlled by training impulse rather than absolute volume. Please see my last specific comment below for further definition of this point.

Conclusion: the conclusion is succinct and based off the obtained results. I reiterate that HIIT and MICT might not be the proper terms here, but this has already been addressed above and below.

Please find specific comments detailed below:

L64-68: What is the difference between high-intensity and vigorous-intensity exercise training? What are the definitions and boundaries adopted by the authors to determine these different exercise domains? This reviewer is more familiar with the moderate-heavy-severe exercise domain continuum, which is defined by objective boundaries.

L68-71: In the methods session the authors further explore this rationale and propose a controlled volume in the current study. I recommend that this rationale (session volume) be presented in the introduction to better conduct the reader to the research problem.

L76-82: Have the authors considered that increased brain lactate uptake during exercise is natural due to lactate being a preferred fuel for astrocytes over glucose? I reckon that overconsumption of lactate by the brain might be related to increased availability (resulting from intense muscle contractions) rather than to increased demand for cognition and/or neurogenesis purposes.

L117: Please revise hypothesis “iii” (there are two “iii” and no “ii”).

L136-140: What dependent variable was used for sample size calculations?

L159-160: Why did the author choose to analyze blood lactate concentration three minutes post-exercise and not upon completion?

L271: Please revise “differences” and include “time vs condition effects” instead.

274-275: If sample size calculations were performed and adhered to, please refrain from presenting “trends”. CSB was not affect by HIIT.

This is a major concern regarding this study. Could it be that the load of the session, rather than the intensity/duration of the exercise, be the factor influencing cognition? Are there studies comparing HIIT and MICT sessions with the same training impulse (TRIMP) (i.e., duration x intensity, which could me measured as physiological responses to exercise such as HR, blood lactate accumulation, etc) and showing different results regarding exercise-induced impacts in cognition? I have estimated the training impulses using both mean absolute HR and blood lactate concentration at 3 min post-exercise and the results were as follows:

HR TRIMP (i.e., mean absolute HR x 14 min) – MICT: 2,136 A.U. vs HIIT: 2,294 A.U.

Blood Lactate TRIMP (i.e., blood lactate concentration at 3 min post-exercise x 14 min) – MICT: 26 A.U. vs HIIT: 143 A.U.

If blood lactate is to be used as the main dependent variable influencing cognition in the study to compare MICT and HIIT training sessions with the same load, MICT sessions would have to be ~77 min long to result in the same training load elicited by HIIT sessions.

I understand that this rationale has limitations, but I also believe that the comparisons in the study are not adequate, since the volume of MICT adopted in the present study, although identical to HIIT, might not have been sufficient to induce the exercise-related benefits of this type of exercise, which is known to depend on longer-duration sessions. In conclusion, perhaps longer MICT sessions achieving the same load could have a similar impact on cognition despite of smaller blood lactate accumulation.

Author Response

Answer to Reviewer 3

The manuscript “Memory modulation by exercise in young adults is related to lactate and not affect by BDNF polymorphism” aimed to investigate the impacts of different-intensity exercises, BDNF polymorphism and sex on memory modulation. I commend the authors for their work. General comments and specific points and sections are provided below:

  • We are grateful for the reviewer´s suggestions, and hopefully, in the comments below, we can resolve all the doubts.

General comments

Presentation: the presentation is good and the writing is sound, with no need for professional revision. I recommend that the authors screen the manuscript for minor typos. Some of the abbreviatures used in the study are not defined (e.g., MAS, FCmax, BMI, etc).

  • MAS, HRmax and BMI have been defined in text. FCmax was a typo.

Novelty: the topic itself is not novel, but the perspective adopted in the study is.

  • Thank you very much for the appreciation.

Title: the title is well written, but does not cite the sex aspect of the study. Please consult other comments on the use of the terms MICT and HIIT.

  • Sex has been included in title.

Abstract: overall, the abstract is well written and provides the essential information. I did miss the description of MICT sessions in the abstract. Also, the opening statement (“HIIT is on the rise compared to MICT due to its benefits for health and performance”) can be misleading as a possible superiority of HIIT over MICT is still under heated debate in the literature.

  • Your recommendation has been duly noted.
  • The description of the MICT protocol is now included in lines 31-32
  • The opening statement has been changed to avoid any confusion: “Currently, high-intensity interval exercise (HIIE) is on the rise compared to moderate-intensity exercise (MIE) due to its similar benefits for health and performance with low time requirements.”

Introduction: the introduction is well written and briefly presents some of the topics the study aims to investigate. However, probably due to the many aims and hypotheses presented, the rationale is relatively shallow, with as little as a couple of lines invested in presenting sex-related differences in BDNF secretion and cognition. The link between lactate and cognition/memory modulation should also be further addressed, as it appears to be the main argument of the study.

  • Regarding sex and BDNF, we have already included this information in lines 115-116 and 118-119 of the introduction:

“BDNF release differs between sexes, but the impact and mechanisms of such differences remain unexplored [27] […]. Moreover, BDNF genotype does not translate into cognitive benefits in females, but the opposite response has been reported in male subjects [29]

  • As for the topic of lactate and cognition in the introduction, we had included in lines 79-84:

“Lactate release has been previously correlated with cognitive performance in humans [13]. Moreover, as El Hayek, Khalifeh, Zibara, Abi Assaad, Emmanuel, Karnib, El-Ghandour, Nasrallah, Bilen, Ibrahim, Younes, Abou Haidar, Barmo, Jabre, Stephan and Sleiman [14] have demonstrated that  lactate enhanced BDNF production in the  hippocampus, and BDNF, in turn, has been associated with hippocampus learning [16] and general memory processing [17]”

Materials and Method: methods are well described and adequate. I do have one important point to address regarding the choice of the HIIT vs MICT protocols. This comment is presented in the discussion topic below.

Results: the results are clearly presented and I commend the authors for their work in presenting such a large amount of data and yet managing to be clear and concise.

Discussion: the discussion is comprehensive and thorough. All important aspects of the study are clearly addressed and contextualized with the current literature. I have one major concern which mostly related to terminology. The authors choose to adopt the terms HIIT and MICT, but the characteristics of the MICT session are not in line with actual moderate intensity continuous training guidelines. MICT should be performed with longer durations to elicit its benefits. The HIIT session appears to be adequate in the study. I suggest that the authors change the terminology in the manuscript to make it clear that what is being compared in the study is the intensity of the exercise (severe vs moderate), rather than the modality (HIIT vs MICT). I would expect that a comparison between HIIT and MICT would be controlled by training impulse rather than absolute volume. Please see my last specific comment below for further definition of this point.

  • Thank you for the advice; we are sure these modifications will improve the article.
  • Regarding MICT, the reviewer´s point of view is more precise than ours. Our moderate exercise session cannot be considered a “MICT” session because of the short duration. Thus, the term “MICT” may be misleading for some readers. To avoid confusion, we have proceeded to substitute the term with “moderate-intensity exercise (MIE)” in the text. To be more consistent with our methodology, that is, we have only evaluated an acute cognitive response pertinent to a single exercise session, we have also changed “HIIT” to “high-intensity interval exercise (HIIE).”

  • Concerning the remark on training impulse, our answer is given below as a response to the final reviewer´s comment.

Conclusion: the conclusion is succinct and based off the obtained results. I reiterate that HIIT and MICT might not be the proper terms here, but this has already been addressed above and below.

Please find specific comments detailed below:

L64-68: What is the difference between high-intensity and vigorous-intensity exercise training? What are the definitions and boundaries adopted by the authors to determine these different exercise domains? This reviewer is more familiar with the moderate-heavy-severe exercise domain continuum, which is defined by objective boundaries.

  • In agreement with Norton et al. (2010) (cited by Oberste et al.,2019), HIIT is different from “vigorous” exercise because vigorous is a continuous training method.

Norton, K., Norton, L., & Sadgrove, D. (2010). Position statement on physical activity and exercise intensity terminology. Journal of science and medicine in sport13(5), 496-502.

Oberste, M., Javelle, F., Sharma, S., Joisten, N., Walzik, D., Bloch, W., & Zimmer, P. (2019). Effects and moderators of acute aerobic exercise on subsequent interference control: a systematic review and meta-analysis. Frontiers in psychology10, 2616.

In the Chang et al. (2012) meta-analysis, high-intensity is defined as a “very-hard” exercise (with efforts >93 %HR). Regarding the continuum domain moderate-heavy-severe related to exercise thresholds (VT1, VT2, Lactate threshold, OBLA, MLSS… or any other method to establish the intensity ranks), vigorous is related to heavy intensity, close to VT2 (or OBLA...), and the Chang´s definition of “high-intensity” should be a severe intensity, above VT2.

The confusion here stems from the common (and unprecise) use of intensities in cognitive-exercise research, based on %HR rather than exercise thresholds. In line with this view, high-intensity exercise translates into vigorous intensity, in agreement with ACSM recommendations (2011), where moderate exercise corresponds to 64-76 %HR and vigorous to 77-95% HR. The mean HR value for the whole duration of our MIE protocol was 77.8 %HR, just in the frontier between moderate and vigorous intensities.

To be more precise, we thought it was important to maintain the authors´ (Chang et al., 2012) point of view. So we have proceeded to include clarifications in the manuscript:

“whenever there is sufficient rest after the exercise bout, very high-intensity (> 93% Heart Rate) exercise would produce the greatest benefits in cognitive function.” (in lines 65-66)

Chang, Y. K., Labban, J. D., Gapin, J. I., & Etnier, J. L. (2012). The effects of acute exercise on cognitive performance: a meta-analysis. Brain research1453, 87-101.

We hope, this answer resolves the doubts.

L68-71: In the methods session the authors further explore this rationale and propose a controlled volume in the current study. I recommend that this rationale (session volume) be presented in the introduction to better conduct the reader to the research problem.

  • We give the reason to limit our protocol to 20 minutes in lines 70-73 of the introduction: “revealed that the lower volume sessions (which, in turn, allow to sustain higher intensi-ties) are associated with better response, with the sessions below twenty minutes, in particular, leading to superior effects [1,3].”

 It is unclear to us which extra information is required concerning the session volume.

L76-82: Have the authors considered that increased brain lactate uptake during exercise is natural due to lactate being a preferred fuel for astrocytes over glucose? I reckon that overconsumption of lactate by the brain might be related to increased availability (resulting from intense muscle contractions) rather than to increased demand for cognition and/or neurogenesis purposes.

  • Lactate is indeed the preferred fuel of astrocytes. Nonetheless, in this article, the cognitive response is related to neuronal function in the hippocampus and the metabolic SIRT1 pathway in neurons which, in turn, activates PGC1 and FNDC5 secretion, a molecule that mediates the BDNF transcription (el Hayek et al., 2019). We cannot be certain that the cognitive response is linked to a stimulation of astrocytes; however, it is related to BDNF production. Moreover, lactate release from astrocytes initiates BDNF production in neighboring neurons, which is relevant for memory formation (Suzuki et al., 2011).

El Hayek, L., Khalifeh, M., Zibara, V., Abi Assaad, R., Emmanuel, N., Karnib, N., ... & Sleiman, S. F. (2019). Lactate mediates the effects of exercise on learning and memory through SIRT1-dependent activation of hippocampal brain-derived neurotrophic factor (BDNF). Journal of Neuroscience39(13), 2369-2382.

Suzuki, A., Stern, S. A., Bozdagi, O., Huntley, G. W., Walker, R. H., Magistretti, P. J., & Alberini, C. M. (2011). Astrocyte-neuron lactate transport is required for long-term memory formation. Cell, 144(5), 810-823. doi:10.1016/j.cell.2011.02.018

L117: Please revise hypothesis “iii” (there are two “iii” and no “ii”).

  • It has been corrected.

L136-140: What dependent variable was used for sample size calculations?

  • Using G-Power, we have decided to assume a medium effect size (ηp2 = 0.05) for all the variables, and it was calculated using ANOVA RM 2x2. The dependent variable is TIME in the ANOVA RM TIME vs Condition.

L159-160: Why did the author choose to analyze blood lactate concentration three minutes post-exercise and not upon completion?

  • It has been previously demonstrated that at least three minutes are required to allow the diffusion of muscle lactate into the capillary blood (Bentley et al. 2007; Pettitt et al. 2013).

Bentley, D. J., Newell, J., & Bishop, D. (2007). Incremental exercise test design and analysis. Sports medicine37(7), 575-586.

Pettitt, R. W., Clark, I. E., Ebner, S. M., Sedgeman, D. T., & Murray, S. R. (2013). Gas exchange threshold and vo2max testing for athletes: an update. The Journal of Strength & Conditioning Research27(2), 549-555.

We believe the confusion generated by figure 2 could give rise to further questions. In reality, the first lactate sample was 3 minutes after the main exercise activity, that means extracted immediately after the cessation of the cooldawn. We understand that the figure may be misleading – the sample was taken after the cooldown and not following three additional minutes. The updated figure has now been included (where “post 3” commentary has been erased) along with the detailed text of the corrected procedure. 

L271: Please revise “differences” and include “time vs condition effects” instead.

  • We are not sure about this advice. In lines 284-286, we have mentioned the lactate differences as a result of a simple T-Test between conditions; there is no Time variable in the analysis.

274-275: If sample size calculations were performed and adhered to, please refrain from presenting “trends”. CSB was not affect by HIIT.

  • “Trend” has been deleted from text and figure.

This is a major concern regarding this study. Could it be that the load of the session, rather than the intensity/duration of the exercise, be the factor influencing cognition? Are there studies comparing HIIT and MICT sessions with the same training impulse (TRIMP) (i.e., duration x intensity, which could me measured as physiological responses to exercise such as HR, blood lactate accumulation, etc) and showing different results regarding exercise-induced impacts in cognition? I have estimated the training impulses using both mean absolute HR and blood lactate concentration at 3 min post-exercise and the results were as follows:

HR TRIMP (i.e., mean absolute HR x 14 min) – MICT: 2,136 A.U. vs HIIT: 2,294 A.U.

Blood Lactate TRIMP (i.e., blood lactate concentration at 3 min post-exercise x 14 min) – MICT: 26 A.U. vs HIIT: 143 A.U.

If blood lactate is to be used as the main dependent variable influencing cognition in the study to compare MICT and HIIT training sessions with the same load, MICT sessions would have to be ~77 min long to result in the same training load elicited by HIIT sessions.

I understand that this rationale has limitations, but I also believe that the comparisons in the study are not adequate, since the volume of MICT adopted in the present study, although identical to HIIT, might not have been sufficient to induce the exercise-related benefits of this type of exercise, which is known to depend on longer-duration sessions. In conclusion, perhaps longer MICT sessions achieving the same load could have a similar impact on cognition despite of smaller blood lactate accumulation.

  • The reviewer is correct, and the two sessions do have different loads. However, there are several reasons why the session duration is constrained in the cognition-exercise research; meanwhile, the use of equivalent loads is avoided.

It is important to remember that our main focus was to optimize the acute cognitive response, and fitness improvement was not considered. In line with this view, Oberste et al. (2019) have suggested in their meta-analysis:

“When the studies were clustered in duration subgroups (duration up to 20 min vs. duration between 21 and 40 min vs. more than 40 min), subgroups with shorter exercise duration apparently yielded more beneficial effects (−0.32 vs. −0.18 vs. −0.07). If all available information (absolute value of exercise duration) was included in the analysis (meta-regression), the positive relationship between exercise duration and magnitude of effects became even clearer.”

(Oberste, M., Javelle, F., Sharma, S., Joisten, N., Walzik, D., Bloch, W., & Zimmer, P. (2019). Effects and moderators of acute aerobic exercise on subsequent interference control: a systematic review and meta-analysis. Frontiers in psychology, 10, 2616.)

This explains why we have also decided to limit the duration of acute exercise and manipulate the load. We are aware that two exercise sessions with different loads would inevitably produce distinct physiological adaptations and fatigue patterns. However, this timeframe has been acknowledged by many as an optimal duration to explore the dose-response relationship as it pertains to the link between cognitive response and intensity (Chang et al., 2015; Chang et al., 2019; Shi et al. 2022). If we are looking to enhance cognitive response, the load should be manipulated, as different exercise intensities stimulate cognitive function to a distinct degree.

It is also worth highlighting that more research is required to analyze the impact of different or equivalent exercise loads on cognitive function. The examination of the dose-response relationship as it concerns intensity-cognitive response link is fundamental, but so are the remaining FITT (frequency, intensity, time, type) variables and the relationship between them (exercise load). Unfortunately, many questions still remain unanswered regarding the impact of the FITT dimensions, and, of course, we are still far from fully elucidating the implications of load in exercise-cognition research.

Round 2

Reviewer 1 Report

Thank you very much for considering the comments and for the effort you are putting into trying to publish the article in this journal.

You have provided interesting information and some supporting quotes, but you acknowledge and accept my earlier comments. These comments imply great physiological limitations that have not been overcome in the revised version, therefore, with great regret, I cannot consider this article as suitable for publication.

Author Response

  • We have already included the updated version of the discussion (line 448-462):

“Moreover, the training load (a product of the intensity and duration of the exercise session) varies between experimental protocols, and we are aware that the magnitude of the training load will determine the physiological adaptations following an exercise bout. The reason for limiting both protocols to 20 minutes is founded in the previous research, where this timeframe has provided the most benefit for cognitive function [3,58,59]. Thus, once we have settled upon the optimal duration, we can only manipulate the exercise intensity, which would inevitably produce different loads, and could be considered a limitation of the present study. But the issue, however, is the lack of previous evidence on the relationship between exercise load and cognitive function.

In future research, the focus should be placed on the examination of the dose-response relationship as it concerns the intensity-cognitive response link, but it is also fundamental to explore the remaining FITT (frequency, intensity, time, type) variables and the relationship between them (exercise load). Many questions still remain unanswered regarding the impact of the FITT dimensions, and, of course, we are still far from fully elucidating the implications of load in exercise-cognition research.”

  • We are quite disappointed to hear this point of view. We have tried to demonstrate the adequate research design assuming the training load limitation to the reviewer. The physiological point of view of the research, correlating lactate with cognitive improvement in agreement with neuroscience evidence in murine models, was the article's objective and was correctly designed to achieve the aim. The nonexistent training load paradigm in the exercise-cognitive research field is a solid reason to avoid the load point of view in the protocol design. Moreover, the arbitrary measure of load (as arbitrary units always measure it) is far away from the neuroscience knowledge about acute and chronic molecular mechanisms on cognitive response related to time, intensity, or fitness, but not to the session load. So we must reaffirm that there is no physiological reason to modify our research design in studying the exercise-cognition dose-response relationship.

Reviewer 3 Report

I thank the authors for their responses and revisions. The minor points I have presented in my first review have been succesfully addressed in the manuscript. The major point I have raised was also addressed by the authors in their response, however, it has not been incorporated to the manuscript (which is the final output of this process and should include the limitations). Also, I would recommend that the authors include their response regarding sample size calculations in the manuscript. I feel that the author's responses were sufficient to address all of my comments and I trust that they will include the limitations mentioned above in the manuscript. Therefore I recommend the acceptance of the manuscript with minor revisions (which do not need to be sent for my appreciation). 

Author Response

Dear Reviewer,

We have included the following paragraphs in the manuscript to attend to your suggestions. Hopefully, we have not missed any of your advice.